# Açaí (*Euterpe oleracea* Mart.) Seed Oil Exerts a Cytotoxic Role over Colorectal Cancer Cells: Insights of Annexin A2 Regulation and Molecular Modeling

**DOI:** 10.3390/metabo13070789

**Published:** 2023-06-25

**Authors:** Marcos Antonio Custódio Neto da Silva, Josiane Weber Tessmann, Kátia Regina Assunção Borges, Laís Araújo Souza Wolff, Fernanda Diniz Botelho, Leandro Alegria Vieira, Jose Andres Morgado-Diaz, Tanos Celmar Costa Franca, Maria do Carmo Lacerda Barbosa, Maria do Desterro Soares Brandão Nascimento, Murilo Ramos Rocha, João Ernesto de Carvalho

**Affiliations:** 1Faculty of Medical Science, Post-graduation in Internal Medicine, State University of Campinas, Campinas 13083-970, Brazil; marcos.antonio@ufma.br; 2Nucleum of Basic and Applied Immunology, Pathology Department, Federal University of Maranhão, São Luís 65080-805, Brazil; kareborges@gmail.com (K.R.A.B.); laiswolff19@gmail.com (L.A.S.W.); carminha13032009@hotmail.com (M.d.C.L.B.); m.desterro.soares@gmail.com (M.d.D.S.B.N.); 3Cell Structure and Dynamics Group, Cellular and Molecular Oncobiology Program, National Cancer Institute, Rio de Janeiro 20231-050, Brazil; jotessmann@gmail.com (J.W.T.);; 4Laboratory of Molecular Modeling Applied to Chemical and Biological Defense (LMCBD), Military Institute of Engineering, Rio de Janeiro 22290-270, Brazil; fernanda.botelho@ime.eb.br (F.D.B.); leandrovieira@ime.eb.br (L.A.V.); tanosfranca@gmail.com (T.C.C.F.); 5Department of Chemistry, Faculty of Science, University of Hradec Kralove, Rokitansheho 62, 500-03 Kralove, Czechia; 6Faculty of Pharmaceutical Sciences, Post-graduation in Internal Medicine, State University of Campinas, Campinas 13083-970, Brazil

**Keywords:** *Euterpe oleracea* Mart., polyphenols, apoptosis, autophagy, Annexin A2

## Abstract

Açaí, *Euterpe oleracea* Mart., is a native plant from the Amazonian and is rich in several phytochemicals with anti-tumor activities. The aim was to analyze the effects of açaí seed oil on colorectal adenocarcinoma (ADC) cells. In vitro analyses were performed on CACO-2, HCT-116, and HT-29 cell lines. The strains were treated with açaí seed oil for 24, 48, and 72 h, and cell viability, death, and morphology were analyzed. Molecular docking was performed to evaluate the interaction between the major compounds in açaí seed oil and Annexin A2. The viability assay showed the cytotoxic effect of the oil in colorectal adenocarcinoma cells. Acai seed oil induced increased apoptosis in CACO-2 and HCT-116 cells and interfered with the cell cycle. Western blotting showed an increased expression of LC3-B, suggestive of autophagy, and Annexin A2, an apoptosis regulatory protein. Molecular docking confirmed the interaction of major fatty acids with Annexin A2, suggesting a role of açaí seed oil in modulating Annexin A2 expression in these cancer cell lines. Our results suggest the anti-tumor potential of açaí seed oil in colorectal adenocarcinoma cells and contribute to the development of an active drug from a known natural product.

## 1. Introduction

Cancer is a growing health problem worldwide due to the increase in life expectancy, urbanization, and subsequent changes in environmental conditions. According to data from GLOBOCAN for 2020, 19.3 million cases of cancer and approximately 10 million deaths were estimated. The most frequent types of cancer were breast cancer (11.7%), lung cancer (11.4%), and colorectal cancer (10.0%). Lung cancer remains the leading cause of death from neoplasia followed by colorectal cancer and hepatocarcinoma [1]. 

In Brazil, estimations for each year of the triennium 2023–2025 indicate that there will be 704,000 new cases of cancer. Non-melanoma skin cancer is the most frequent (220,000) followed by cancers of the breast and prostate (74,000 and 72,000, respectively), colon and rectum (46,000), lung (32,000), and stomach (21,000) [2].

Two of the major challenges in treating neoplasias are the inherent toxicity and side effects of anti-tumoral drugs and the development of resistance to these treatments. Hence, there is a constant search for adjuvant treatments that might increase the efficacy of traditional chemotherapeutic drugs and hinder the development of resistance. The immense biodiversity present in the Amazon Forest and the popular knowledge of its inhabitants provide fertile grounds for the discovery of new and promising neoadjuvant agents.

Açaí is considered a food of high caloric and nutritional value with a high percentage of lipids, proteins, and minerals and is the main food base of the riverside population of the Amazon River Estuary region [3]. Previous phytochemical analyses have revealed the presence of flavonoids, anthocyanins, benzenoid lignans, benzoquinone, monoterpenoids, norisoprenoids, and essential fatty acids [4,5,6,7]. Mantovani et al. (2003) described the predominance of unsaturated fatty acids, especially oleic acid and palmitoleic acid [8].

“Açaí” is a term from the *tupi* origin “yasa’y” (i), which means “water palm tree” [9,10]. The fruits of the açaí tree are extracted wine, pulp, or simply açaí, as is known in the region. Açaí is usually eaten with *mandioca* flour, which is associated with fish, shrimp, or beef, and is the basic food for riparian communities. 

In addition to serving as a food, açaí is widely used in folk medicine [11,12]. In ethnomedicine, the root and stem of the leaves are used for muscular pain and snake bites and to relieve chest pain [13,14]. The root can also be used for the treatment of malaria and liver and kidney infections [15,16]. The seed provides a dark green oil popularly used as an antidiarrheal [15]. In 2002, a study using *Euterpe oleracea* leaves reported a reduction in abdominal contortions and peripheral analgesic action [17]. 

Some comprehensive studies have shown different biological activities of açaí, such as anti-lipidemic, neuroprotective, hypocholesterolemic, therapeutic, anti-inflammatory, and anti-cancer properties [18,19,20,21,22].

Regarding açaí oil, Melhorança Filho and Pereira (2012) and Magalhães et al. (2020) described antibacterial activity [23,24], Favacho et al. (2011) described anti-inflammatory and antinociceptive effects [25], and Souza et al. (2017) described antilipemic action [26].

Marques et al. (2017) studied açaí oil and different human cells, evaluating the cytotoxicity, genotoxicity, and antigenotoxicity of *Euterpe oleracea*. HepG2 cells (hepatoma cells) and human leukocytes were used in this study. No cytotoxic effects of the extracts were observed on the strains used [27].

Regarding its anti-tumoral effect, despite the bioactive potential of açaí, only a few studies have been described in the literature with promising results showing the chemopreventive and therapeutic effects of açaí in different cancer models, including esophageal cancer [28], urothelial cancer [29], colon cancer [30,31,32,33], melanoma [34], and breast cancer [35,36,37,38]. 

Annexin A2 protein has been investigated as a prognostic marker because of its wide presentation in various forms of cancer. Deletion of the Annexin A2 gene (ANXA2) has been shown to decrease DNA synthesis and cell proliferation, suggesting that Annexin A2 is a factor in cell division [39]. 

Based on the current knowledge about the use of natural products with antioxidant, anti-inflammatory, and anti-tumor properties, our goal was to analyze the effects of açaí seed extract and oil (*Euterpe oleracea* Mart.) in different cell lines of human colon adenocarcinoma (CACO-2, HT-29, and HCT-116), providing evidence that suggests its use in a neoadjuvant setting after preclinical studies.

## 2. Materials and Methods

### 2.1. Materials

The following antibodies were purchased from commercial sources: anti-Bax (Cell Signaling Technology, Inc., Danvers, MA, USA), anti-Bcl-2 (Cell Signaling Technology, Inc.), total anti-Annexin A2 (Invitrogen), phospho-anti-Annexin A2 (R&D Systems), and anti-LC3B (Cell Signaling Technology).

### 2.2. Preparation of Lyophilized Hydroalcoholic Seed Extract and Oil of Euterpe Oleracea Mart

The fruits of açaí (*Euterpe oleracea* Mart.) used in this study were obtained from Juçara Park (São Luís, Maranhão, Brazil). A sample of the specimen was stored under exsiccate number 30 issued by the Rosa Mochel Herbarium of the Nucleus of Biological Studies of the State University of Maranhão (UEMA) and deposited with the World International Property Organization under registration number PI0418614-1.

The fruits were previously refrigerated at −20 °C in the Laboratory of Cell Culture of the Nucleus of Basic and Applied Immunology of the Federal University of Maranhão (UFMA). After thawing at room temperature, the samples were separated into three parts: the seed, pulp, and total fruit (seed + pulp). The extraction process followed the methodology developed by de Moura et al. (2012) [40].

Approximately 360 g of açaí was washed with tap water and boiled in distilled water for 5–10 min. Subsequently, the portions were ground and homogenized with 400 mL of ethanol under stirring for 2 h. The resulting extracts were stored at 4 °C and protected from light for 10 d. After this maturation period, the hydroalcoholic extracts were filtered through Whatman #1 filter paper, and the liquid phase was concentrated in a low-pressure rotary evaporator (Fisatom Equipamentos Científicos Ltd.a., São Paulo, Brazil) at approximately 40 °C and then lyophilized (LIOTOP model 202, Fisatom Equipamentos Científicos Ltd.a., São Paulo, Brazil) at a temperature of −30 to −40 °C and a vacuum of 200 mm Hg. The extracts were kept at −20 °C until the day of use.

For oil extraction, the fruits were washed under running water and subsequently subjected to pulping. After this process, 360 g of crushed seeds of *Euterpe oleracea* Mart. was dried in the sun and crushed in a mill. After the aforementioned pre-treatment, the oil was extracted using a Soxhlet extractor. The solvent used in the process was n-hexane, and the total extraction time was 6 h.

### 2.3. MS/MS Analysis

Crude extracts of the seeds, pulp, and total fruit of açaí (*Euterpe oleracea* Mart.) were suspended in 2 mL of MeOH HPLC and centrifuged at 13,000 rpm for 5 min. The supernatant solution (100 µL) was filtered through 0.22 µm and diluted into 900 µL MeOH HPLC. The samples were analyzed using an LC Agilent 1200 mass spectrometer coupled with an Agilent iFunnel 6550 Q-ToF LC/MS. The electrospray ionization source operated in positive mode ESI (+) following operating conditions: nebulizing gas temperature, 290 °C; capillary voltage, +3500 V; nozzle voltage, 320 V; drying gas flow, 12 mL min^−1^; nebulization gas pressure, 50 psig; auxiliary gas temperature, 350 °C; and flow of auxiliary gas, 12 mL min^−1^. The analyzer’s time-of-flight (ToF) was operated in the range of *m*/*z* 50–1500. Collision energy formula (auto MS/MS mode): 4 V (slope) × (*m*/*z*)/100 + 5 V (offset). A maximum of five precursors per cycle were selected. Stationary phase: Thermo Scientific Accucore C18 2.6 µm, 2.1 mm × 100 mm. Mobile phase: acetonitrile and 0.1% formic acid. Flow rate: 0.2 mL/min^−1^. The organic phase was run in gradient mode from 5% to 98% within 10 min, held for 5 min, up to 5% within 1.2 min, and held for 4.8 min. Total run time: 20 min. The injection volume used was 2 µL. The spectra were processed using Agilent Mass Hunter Workstation Software.

### 2.4. Oil Esterification

The crude oil of *Euterpe oleracea* Mart. was analyzed in the form of methyl esters prepared according to Hartman and Lago (1973) [41]. The oil (100 mg) was weighed and placed in 20 mL tubes with a screw cap. Then, 4 mL of a 0.5 mol/L solution of sodium hydroxide in methanol was added. The mixture was heated for a period of 5 min in a boiling water bath until the fat globules completely dissolved, and the tubes were quickly cooled in running water, immediately adding 5 mL of a solution previously prepared with 1 g of ammonium chloride, dissolved in 30 mL of methanol and 1.5 mL of sulfuric acid, and concentrated in small portions with stirring. Afterward, the tube was shaken, heated in a boiling water bath for 5 min, cooled under running water, and 4 mL of saturated sodium chloride solution was added and stirred for 30 s. Finally, 5 mL of hexane was added, and the tube was shaken vigorously using a vortex mixer for 30 s and left to rest for complete separation of the phases, which were kept cooled for chromatographic analysis.

### 2.5. GC-MS Analysis

Fatty acid identification was performed using a gas chromatograph (GC-2010) coupled to a mass spectrometer (GC-EM QP2010 Plus; Shimadzu, Kyoto, Japan). For chromatographic analysis, a capillary column ZB-FFAP (30 m × 0.25 mm × 0.25µm) was used for the chromatographic analysis. The flow of carrier gas was helium at a linear speed of 30 cm/s and column flow of 1.0 mL / min. The oven program was 120 °C for 2 min with a heating ramp of 10 °C/min up to 180 °C that remained for 5 min, then heated again at a rate of 5 °C/min up to 230 °C, remaining for 25 min. The temperatures of the injector and the ion source were 200 °C and 250 °C, respectively. Split injection mode at a ratio of 50. 

The quantification of fatty acids was performed by normalizing the peak areas, and the identification of esters from the fatty acids that make up the oil was performed using the NIST08 equipment library (National Institute of Standards and Technology).

### 2.6. Cell Culture

Human colorectal adenocarcinoma cell lines HT-29 (HTB-38TM), HCT-116 (ATCC^®^ CCL-247^™^), and Caco-2 (HTB-37TM) were obtained from the American Type Culture Collection (Manassas, VA, USA). The cells were grown in Dulbecco’s modified Eagle’s medium (Invitrogen Inc, Carlsbad, CA, USA) supplemented with 10% fetal bovine serum (FBS), penicillin G (60 mg/L), and streptomycin (100 mg/L) at 37 °C in a humidified atmosphere of 5% CO_2_/air, and the cells were passaged weekly by using a solution of 0.05% trypsin/0.02% EDTA in PBS. For experiments, cells were seeded into culture flasks, plates, or glass coverslips.

### 2.7. Treatments with Euterpe Oleracea Mart Seed Extract and Oil

Samples were diluted in dimethyl sulfoxide (DMSO) stock solutions (Merck) at a concentration of 0.1 g/mL. Cells were seeded in 96-well plates at a concentration of (1 × 10^4^ cells/mL) and after 24 h were treated with 0.25, 2.5, 25, or 100 μg/mL açaí seed extract and oil. The cell viability was assessed at 24, 48, and 72 h.

### 2.8. Cell Viability Test—MTT

Cells were trypsinized and counted in a Neubauer chamber, and an aliquot of 1 × 10^4^ cells/mL were cultured in 96-well plates in the presence or absence of the extract and oil. After 24 and 48 h of treatment, 200 μL of fresh medium containing 10 μL of 3-(4,5-Dimethylthiazol-2-yl)-2,5-diphenyltetrazolium bromide (MTT) was added to the culture. The cells were re-incubated in a CO_2_ incubator for 3 h and protected from light. Subsequently, the plates were centrifuged at 1200 rpm for 5 min at 4 °C. The supernatant was discarded, and 100 μL of dimethyl sulfoxide (DMSO) was added to each well. The absorbance was measured on a Spectra Max 190 plate spectrophotometer (Molecular Devices, Sunnyvale, CA, USA) at a wavelength of 538 nm.

In light of the initial results and to evaluate the mechanism of action of açaí seed oil, CACO-2, HCT-116, and HT-29 cells were pre-treated with N-acetylcysteine (NAc), a known antioxidant, at doses of 2.5, 5, and 10 mM for 2 h before treatment with açaí seed oil at concentrations of 25 and 50 μg/mL for 24 h. Cells were quantified using Trypan blue to assess the percentage of viable cells, and then a cell viability assay was performed using MTT.

### 2.9. Morphological Analysis via Inverted Light Microscopy 

Cell morphology was analyzed via light microscopy after treatment with açaí seed extract, and oil was analyzed using an inverted Axio Observer Z1 microscope equipped with an Axiocam HRc Ver.3 chamber. Image analysis was performed using Axiovision Release 4.8.1 software (Carl Zeiss Inc., Jena, Germany). The cells were cultured in 12-well plates in the presence or absence of açaí seed oil for 24 and 48 h and then observed under a microscope.

### 2.10. Annexin—V Assay

After a 24 h incubation period, the cell suspensions were centrifuged at 1000 rpm (168× *g*) for 10 min and resuspended in PBS (pH 7.4). Then, the cells were centrifuged again at 1000 rpm (168× *g*) for 10 min.

Subsequently, viable, apoptotic, and nonviable cells were determined using an ANNEXIN V-FITC apoptosis detection kit containing Annexin V-FITC, propidium iodide, and buffer (BD Biosciences).

After the second centrifugation, the cells were resuspended in binding buffer previously diluted in deionized water at a ratio of 1:10. Then, 500 μL of the cell suspension was labeled with 5 μL Annexin V-FITC and 10 μL propidium iodide.

After 10 min of rest in the dark, cell fluorescence was immediately determined using a flow cytometer. Cells in the early stages of apoptosis were marked intensely with Annexin V-FITC, which emits green fluorescence as a result of its preferential binding to phosphatidylserine residues, externalized at the beginning of the process. Necrotic or non-viable cells were marked intensely with propidium iodide, which emits red fluorescence, and less intensely with Annexin V-FITC. Viable cells were not labeled with Annexin V-FITC or propidium iodide.

Excitation was performed using an argon laser operating at 488 nm, and fluorescence detection was performed at 530 nm (PI) and 670 nm (7-AAD).

### 2.11. Western Blotting

Western blotting was performed as described by Albuquerque-Xavier et al. (2012) [42]. The cells were treated for 12 h with açaí seed oil, and the protein content was extracted. Total cell lysate was obtained by incubating cells with lysis buffer (1% Triton X-100, 0.5% sodium deoxycholate, 0.2% SDS, 150 mM NaCl, 2 mM EDTA, 10 mM HEPES (pH 7.4) containing 20 mM NaF, 1 mM orthovanadate, and protease inhibitor cocktail (1:100 dilution)) for 30 min at 4 °C, and centrifuged at 10,000× *g* for 10 min at 4 °C. The supernatant was removed and stored at −20 °C until further use.

Protein quantification was performed using the BCA kit (Bio-Rad, Hercules, CA, USA). Proteins (40 μg/mL) were electrophoretically separated with SDS-PAGE on 13% gels and transferred to a nitrocellulose membrane (Bio-Rad) for 1 h at 10 V. After blocking with 5% milk for 1 h, the membrane was incubated overnight at 4°C with anti-LC3B antibody (1:1500), anti-Bax (1:250), anti-Bcl-2 (1:1500), and total and phosphorylated anti-Annexin A2 (1:1400). Subsequently, the membrane was washed with TBS-T buffer (20 mM Tris-HCl pH 7.6, 137 mM NaCl, and 0.1% Tween-20) and incubated for 1 h with HRP-conjugated anti-rabbit IgG secondary antibody (1:10,000).

The proteins were visualized using a chemiluminescence kit (Amersham Biosciences, Buckinghamshire, UK). All membranes were reused for GADPH labeling to confirm the application of the same amount of protein in all wells. The intensity of the bands was quantified according to their thickness using LabWorks 4.6 program (Bio-Rad Laboratories, Hercules, CA, USA).

### 2.12. Molecular Modeling Studies

#### 2.12.1. Obtention of the 3D Structure of Annexin A2 and Construction of the Ligands

The 3D structure of Annexin A2 (ANXA2) in complex with a tetrasaccharide derived from heparin was downloaded from the Protein Data Bank website (www.rcsb.org, accessed on 18 March 2021) under the PDB ID:2HYU [43]. This structure was used for modeling ANXA2 interactions with the five most abundant fatty acids in the extract.

The 3D structures of the five fatty acids investigated in this work (palmitic, myristic, lauric, oleic, and linoleic) were constructed using software Spartan 8 [44] and optimized using the semi-empirical method PM3 [45] with the atomic partial charges calculated using the natural population analysis method [46]. The molecules were then transferred (together with the ANXA2 structure downloaded from the PDB (www.rcsb.org, accessed on 18 March 2021) to the Molegro Virtual Docker software (MVD^®^) to run the docking studies. 

#### 2.12.2. Docking Studies

To verify the protein’s preferential region for the docking of fatty acids, they were submitted to blind docking over a search spaced with the whole protein structure where the best 100 poses of each ligand were collected. Subsequently, further docking runs were performed, this time with the search spaces restricted to the two regions that concentrated the larger number of poses during the blind docking. For each docking, 6 runs were performed with the 30 best poses collected after each run. These poses were analyzed according to their Moldock scores and positions on the surface of the protein. The most representative positions of each ligand in both regions were selected for further MD simulation studies. The protocols used to perform all docking studies were the same as those previously validated and used [47,48,49].

#### 2.12.3. Molecular Dynamics Simulations

The complexes between ANXA2 and the best poses obtained from the docking studies were subjected to additional MD simulations using the software GROMACS 2019.4 [50,51] and the force field OPLS/AA [52,53]. Each pose was first submitted to the Open Babel [54] and Antechamber Python Parser Interface (ACPYPE) [55] software to generate their coordinates (. growth extension) and topology (. top extension) files, which must be recognized by GROMACS 2019.4 [50,51]. The coordinate and topology files of ANXA2 were generated using the routine pdb2gmx of GROMACS 2019.4 [50,51] with further selection of the force field OPLS/AA [52,53].

Each protein–ligand complex was centered in a cubic box of 929 nm^3^ with a minimal distance solute-box wall (1.5 nm) containing approximately 28,000 TIP3P [55] water molecules under periodic boundary conditions (PBC). Next, the complexes were subjected to two steps of energy minimization using the *steepest descent* algorithm with and without position restraints (PR) of the protein and ligand and convergence criteria of 100.00 kJ mol^−1^ nm^−1^. After energy minimization, 2 equilibration steps meant to bring the system to physiologic conditions of T = 310 K and P = 1 bar were performed. The first step ran under constant T and V (NVT), whereas the second ran under constant T and P (NPT). The stabilities of T and P were maintained using the thermostat *Velocity-rescale* [56] and Parrinello and Rahman’s (1981) pressure coupling methods [57], respectively.

The production step consisted of 50 ns of free MD simulation for each system at 310 K and 1 bar with an integration time of 2 fs and a cut-off of 1.2 nm for VDW and electrostatic interactions. The coordinates and energy data of the complexes were stored after each 10 ps of simulation to enable further analysis of temporal properties, such as total energy, root mean square deviation (RMSD), and average number of H-bonds. All analyses were performed using the software xmgrace 5.1.25 [58] and Visual Molecular Dynamics 1.9.3 (VMD) [59]. 

### 2.13. Statistical Analysis

For statistical analysis of the experimental data, one- or two-way analysis of variance (ANOVA) tests were performed followed by Dunnett’s or Tukey’s post-hoc tests according to the type of analysis. Differences were considered statistically significant at *p* < 0.05. Statistical analysis was performed using GraphPad Prism version 8.4.0 for Windows (GraphPad Software, San Diego, CA, USA).

## 3. Results

### 3.1. Açaí Seed Extract Is Rich in Flavonoids 

For *Euterpe oleracea* Mart. seed extract, 13 compounds were identified via LC-MS (Appendix A), especially flavonoids and anthocyanins. The major compounds were epicatechin, kaempferol-3-O-rutinoside (Figure 1A), nobiletin, dihydrokaempferol, diosmetin, 3-O-Methylquercetin or isorhamnetin, isoorientin, 3-Genistein-8-C-glucoside, and apigenin 6,8-digalactoside.

The yield of the oil obtained in 360 g of seed was approximately 5.6%. The oil was composed of 49.3% of saturated fatty acids, 50.7% unsaturated fatty acids, 29.7% monounsaturated, and 21% polyunsaturated. The major methyl esters were oleic, linoleic, myristic, and palmitic acid. However, other esters appeared in smaller amounts from capric, palmitoleic, linolenic, stearic, eicosanoic, behenic fatty acids, and tricosanoic acid (Table 1 and Figure 1B).

### 3.2. Açaí Seed Oil but Not Extract Exerts Cytotoxic Effect on Colorectal Cancer Cell Lines

The cytotoxic effect of *E. oleracea* Mart. seed oil (in the concentrations 0.25, 2.5, 25, and 100 μg/mL) was analyzed with an MTT (3-(4,5-Dimethylthiazol-2-yl)-2,5-diphenyltetrazolium bromide) assay in three colon adenocarcinoma cell lines: Caco-2, HCT-116, and HT-29.

There was no cytotoxic effect of açaí seed extract on colorectal adenocarcinoma cell lines (Figure 2A). For the açaí seed oil, there was a reduction in cell viability after 24 h at concentrations of 25 and 100 μg/mL in the cell lines Caco-2 (*p* < 0.05) and HCT-116 (trend). In the HT-29 cell line, there was a trend of reduction in cell viability only at the concentration of 100 μg/mL (Figure 2B). After 48 h, treatment with 25 μg/mL seemed not to disturb the typical morphology in the HT-29 cells but lead to noticeable damage and cell death in the other 2 cell lines (Figure 2C).

The IC50 was calculated after 24 h of treatment with açaí seed oil. The most sensitive strain was HCT-116 with an IC50 of 11.8 μg/mL, and the most resistant was HT-29 with an IC50 of 51.2 μg/mL (Figure 2D).

### 3.3. Açaí Seed Oil Induces Cellular Death through Increased ROS

To assess whether the reduction in cell viability occurred due to apoptotic cellular death, an Annexin V/PI assay was performed. Figure 3A shows that treatment with 25 μg/mL of açaí seed oil increased the percentage of cells in the early and late stages of apoptosis in the lines Caco-2 and HCT-116. There was no induction of apoptosis for HT-29 at the analyzed concentration.

Then, cell viability was initially evaluated with manual counting of dead and alive cells after treatment with açaí seed oil and in cells pre-treated for 2 h with 5 mM of n-acetylcysteine (NAc) to evaluate if the reduction in cell viability occurred due to an increase in free radicals.

For the Caco-2 line, cells pre-treated with NAc showed a slight increase in the percentage of live cells when compared to cells treated with oil at concentrations of 25 and 100 μg/mL. In HCT-116 cells, there was an increase in cell survival when pre-treated with NAc + 25 μg/mL oil. For the concentration of 100 μg/mL, there was no increase when pre-treated with NAc, as this concentration is very cytotoxic for this cell line. For the HT-29 lineage, there was a significant increase in tumor cell survival when pre-treated with NAc + 100 μg/mL oil (Figure 3B).

To confirm the results obtained by the quantification of dead and live cells with the Trypan blue stain, a cell viability assay was performed with MTT. All cell lines were pre-treated with NAc at concentrations ranging from 2.5 to 10 mM for 2 h and then treated with açaí seed oil at concentrations of 25 and 50 μg/mL for 24 h.

For the Caco-2 cell line, pretreatment with NAc did not increase cell viability at the three concentrations analyzed when compared to treatment with açaí seed oil at the concentrations tested. HCT-116 cells pre-treated with 10 mM NAc and exposed to 50 ug of açaí seed oil exhibited an increase of cellular viability when compared to control (no pre-treatment. The same pattern was observed for HT-29 cells (Figure 3C).

### 3.4. Autophagy and Annexin A2 Seem to Participate in Cellular Response to Açaí Seed Oil

Considering the role of reactive oxygen species in the death induced via açaí seed oil treatment, the autophagic process was analyzed through the presence of the LC3-B II protein. During autophagy, LC3B-I protein is converted into LC3B-II, and therefore the ratio LC3BII/LC3B-I is related to an increase in autophagosome formation [60]. Annexins are a family of calcium-dependent phospholipid-binding proteins involved in membrane trafficking and organization. Annexin A2 (ANXA2), one of the twelve human Annexins, has been linked to a variety of tumors, including colorectal cancer (CRC). Due to its phospholipid binding properties and relation to autophagy and oxidative stress, the ANXA2 level was also assessed via western blotting.

The protocol for this analysis was comprised of a 5 mM NAc 2 h pre-treatment and then incubation with 25 ug/mL açaí seed oil in a culture media with its respective controls for the 3 cell lines. A decrease in the LC3B-II/LC3B-I protein level was observed in Caco-2 and a greater magnitude in HCT-116 cells when compared to HT-29. This might explain their enhanced sensitivity to the ROS-mediated cytotoxic effects of the açaí seed oil. Annexin A2 levels go in the opposite direction with higher values of its heavier isoform in HCT-116. It is interesting to observe that the lighter bands are indicative of apoptosis. The cell cleaves ANXA2, forming a truncated shape when the apoptotic process is undergoing. These bands were seen in Caco-2 and were more intense in HCT116, which corroborates previous results. Moreover, the pre-treatment showed a reduction in this truncated form, confirming their effectiveness in preventing ROS-mediated apoptotic induction from açaí seed oil treatment (Figure 4).

### 3.5. Docking

The search space involving the whole ANXA2 structure used for the first docking run consisted of a sphere with radio = 46 Å centered at coordinates x = −11.99; y = 1.01; and z = 214.91. This run showed that most of the poses of the five fatty acids concentrated in the same two regions of ANXA2 circled in Figure 5A. Those regions were therefore defined as the preferential binding regions of those molecules over ANAX2, and 2 new search spaces were then defined over them, both consisting of spheres with radio = 20 Å but coordinates centered, respectively, at x = −10.04; y = −7.17; and z = 224.46 (Region 1 in Figure 5A) and x = −5.62; y = −14.21; and z = 198.75 (Region 2 in Figure 5A). It is important to notice that Region 2 is also where the protein S100A10 binds to ANXA2, forming an heterotetrametric complex containing two units of ANXA2 and two units of S100A10 [60]. There are some literature reports arguing that this tetramer might have an important role in the development and multiplication of cancer cells [61,62,63]. Therefore, the binding of the fatty acids to this region might avoid the formation of the tetramer, consequently reducing the rate of the multiplication of cancer cells.

As mentioned in the methodology, one pose of each fatty acid on each region circled in Figure 5 was selected for additional steps of MD simulation. Besides being representative of that ligand, i.e., having many other similar poses in that region from the blind docking, the poses chosen were the ones showing the lowest Moldock score. This suggests a higher probability of the ligand adopting that conformation after complexing with ANXA2. The energy values obtained for the selected poses, as well as the residues observed forming H-bonds during the docking runs, are listed in Table 2, while the respective poses are illustrated in Figure 5B. In Table 2, it is possible to see that all poses show negative energy values ranging from −84.00 to −112 Kcal.mol^−1^. This suggests that the fatty acids have a good affinity for ANXA2 and are capable of complexing in Regions 1 and 2. 

It is important to notice that the two poses selected for myristic acid are very similar and fall both in the interface between Regions 1 and 2. The difference observed between them is the fact that in one case, the carboxyl group is inside the protein, while in the other, it is pointing outside. 

### 3.6. Molecular Dynamics Simulations

Each of the poses illustrated in Figure 5 were submitted to steps of MD simulations as described in the methodology in order to evaluate its dynamic behavior and stability in the respective binding pockets over time. Pre-analysis of the results showed that the poses in Region 2 presented a more stable dynamic behavior than the poses in Region 1 (data not shown). This result associated with the literature finding that Region 2 is important for the binding between ANXA2 and S100A10 [61,62,63] prompted us to focus our analysis only on the dynamic results of Region 2. 

Figure 6A shows the plots of total energy (average and standard deviation) of the complexes ANXA2–ligand during the MD simulations. As can be seen, all complexes presented energy < −9.5 × 105 kJ.mol^−1^ and low values of standard deviation. This means that they achieved stability during the MD simulation, corroborating the docking studies. 

The values of the RMSD for the complexes ANXA2–ligand are shown in Figure 6B,C. As expected, due to its larger size and mobility, ANXA2 presented RMSD values higher than the ligand in all cases, never passing 8 Å versus 3 Å for the ligands. These results suggest the stabilization of the ligands inside the binding pocket. The higher average value of the RMSD observed for the complex ANXA2-linoleic acid (Figure 6C) points to this ligand as the less promising binder to Region 2 of ANXA2 compared to the others, while the acids myristic and palmitic sound like the most promising.

The plots of the H-bonds formed between the ligands and ANXA2 during the MD simulations (Figure 6D) show that no ligand was capable of forming more than three residues during the simulated time. This was already expected once the ligands were fatty acids and therefore bring only the carbonyl group in their structures capable of forming H-bonds. The best results were observed for the lauric and palmitic acids which formed up to three H-bonds with residues Asn65 and Arg68 during the whole MD simulation. The third best was myristic acid, which formed H-bonds with Lys266 and Lys322 during most of the simulation, and the worst results were of linoleic and oleic acids, which were not capable of maintaining H-bonds during most of the simulation time.

## 4. Discussion

The main flavonoids found in açaí were quercetin, orientin, and their derivatives, as well as proanthocyanidins [4,64,65]. Orientin, homoorientin, vitexin, luteolin, cryoseriol, quercetin, and dihydrokaempferol have also been identified in açaí fruits [4,66]. 

This is the first report of the presence of nobiletin and genistein-8-C-glycoside in açaí (*Euterpe oleracea* Mart.). 

In relation to açaí seed oil, the main fatty acids were myristic acid, palmitic acid, oleic acid, linoleic acid, and lauric acid. Our results show that the fixed oil extracted from the açaí seed is composed of 49.27% of saturated fatty acids and 50.73% of unsaturated fatty acids, with 29.73% monounsaturated and 20.85% polyunsaturated. Mantovani, Fernandes, and Menezes (2003) described a predominance of unsaturated fatty acids, mainly oleic acid with 45.1%; 45.7%; and 45.5% for pericarp, endocarp, and whole fruit, respectively, followed by palmitoleic acid to a lesser extent (4.2%; 4.8%; and 4.3% for pericarp, endocarp, and whole fruit, making up more than 50% of the total fatty acids) [8]. ɣ-linolenic acid, linoleic acid, palmitic acid, and oleic acid were the main constituents described by Mulabagal and Calderón (2012) [67]. Similar results were described by Yuyama et al. (2011) [68] and Nascimento et al. (2008) [69].

Antioxidants, both natural and synthetic, are increasingly used as functional food additives and nutritional supplements. Oxidative reactions can damage proteins, lipids, and DNA due to an uneven ratio of antioxidants to free radicals. The capacity of antioxidants to stop oxidative deterioration in food and pharmaceutical items, as well as in the body and against disease processes brought on by oxidative stress, has heightened interest in them. Numerous examinations exploring the connection between food and human health have been carried out using diverse study focus samples. [70,71].

Açaí seed oil induced apoptosis and morphological changes in the CACO-2 and HCT-116 strains. To investigate the mechanisms associated with cell death, the cells were pre-treated with NAC with an increase in cell viability. Monge-Fuentes et al. (2017), using açaí oil in nanoemulsion, found phototoxicity in melanoma cells with reduced cell viability after treatment. In addition, the morphology of B16F10 cells showed loss of cell volume, presence of apoptotic bodies, and loss of cell adhesion [34]. In the Annexin V and PI assay, cell death occurred via late apoptosis/necrosis, suggesting a photosensitizing effect with reduced proliferation of melanoma cells.

Essential oil of *Myrica gale* L., a plant native to Canada used in traditional medicine, has strong cytotoxic effects on human lung (A549) and colon (DLD-1) cancer cell lines [72], and the essential oil of *Ocimum basilicum* L. and *Psidium guajava* L., Thai medicinal plants, inhibit the proliferation of murine leukemia (P388) and oral squamous cell carcinoma cell lines, respectively [73]. Furthermore, a recent study demonstrates that pine essential oil inhibits growth and induces apoptosis in human liver carcinoma cells by downregulating Bcl-2 expression and telomerase activity [74]. Treatment of YD-8 cells (oral cancer) with essential oil from *P. densiflora* leaves strongly inhibited proliferation and survival and induced apoptosis. Treatment led to the generation of ROS, activation of caspase-9, cleavage of PARP, downregulation of Bcl-2, and phosphorylation of ERK-1/2 and JNK-1/2 in YD-8 cells [75].

Açaí oil contains 3.4 times more phenolic acid and 2–14 times less monomeric and dimeric flavanol compared to pulp extract. Furthermore, both the extract and the oil promoted an increase in the production of reactive oxygen species (ROS) at low concentrations [76]. Marques et al. (2017) studied açaí oil and different human cells, evaluating the cytotoxicity, genotoxicity, and antigenotoxicity of *Euterpe oleracea*. For the study, HepG2 cells (hepatoma) and human leukocytes were used. There was no cytotoxic effect in the strains used [27].

Finally, Dias et al. (2014) evaluated the potential pro-apoptotic effect of açaí-derived polyphenols on HT-29 and SW-480 colorectal adenocarcinoma cells. The results showed that the açaí polyphenolic extract at concentrations from 5 to 20 mg/L inhibited the growth of SW-480 and HT-29 cells with greater reductions for SW-480 cells, also reducing the production of reactive species of oxygen (ROS) [33].

There was an increase in the expression of LC3-B in cells treated with seed oil and a reduction in the expression of LC3-B when pre-treated with NAC. In the evaluation of Annexin expression in cells treated with açaí seed oil, the bands under the Annexin were indicative of apoptosis. The cell cleaved ANXA2, forming a truncated shape. The CACO-2 strain showed more isoforms, which corroborated the Annexin apoptosis assay.

Autophagy is a process of autodigestion aimed at recycling damaged cellular components and organelles in response to various stressful conditions [77]. In tumor cells, autophagy plays dual roles in tumor promotion and suppression [78].

The close interaction between ROS and autophagy is reflected in two ways: the induction of autophagy via oxidative stress and the reduction of ROS via autophagy [79].

Induction of autophagy after nutrient starvation requires the production of hydrogen peroxide that oxidizes autophagy-related protein (ATG) 4. Of the ATG proteins, ATG4 is the only protease that regulates autophagy by processing and deconjugating ATG8 [80]. The oxidation modification mainly inactivates the delipidating activity of ATG4, leading to the increased formation of autophagosomes associated with light chain 3 [81,82].

In addition to the above, which is considered a direct mechanism, an indirect induction of autophagy by ROS can also occur. Adenosine monophosphate (AMP)-activated protein kinase (AMPK), which can suppress the activity of the mammalian target of rapamycin (mTOR), is activated by ROS and leads to the induction of autophagy [83].

Annexin A2 is overexpressed in clear cell renal carcinoma, breast, cervical, colorectal, endometrial, hepatocellular carcinoma, lung cancer, ovarian cancer, pancreatic duct adenocarcinoma, glioblastoma, urothelial carcinoma, acute lymphoblastic leukemia, acute promyelocytic leukemia, and multiple myeloma [84,85,86,87,88,89,90,91,92,93,94,95].

Regarding molecular docking and molecular dynamics, the five fatty acids studied presented theoretical results that suggest the formation of stable complexes with ANXA2. Additionally, these acids bind preferentially to the region close to the N-terminal moiety of this protein where the complexation with S100A10, which is suspected of triggering the growing and proliferation of cancer cells, occurs. Therefore, our results suggest that these acids have the potential of impairing the formation of the ANXA2-S100A10 complex, contributing to the interruption or slowing of the proliferation of cancer cells.

In studies that investigated Annexin A2 using immunohistochemistry, high expression of Annexin A2 was significantly correlated with tumor size, poorly differentiated tumors, depth of invasion, and TNM stage in colorectal cancer [95]. Annexin A2 has been shown to be an independent factor of poor prognosis in patients with colorectal cancer [96]. Annexin A2 in the cell membrane is a hallmark of highly invasive tumors. This ability to invade tissue shows how Annexin A2 can affect lymph node metastasis [86,90]. Annexin A2 has also been shown to be important for the effect of progastrins and gastrins, partially mediating the effect of growth factors on colon cancer cells [97].

Annexin collaborates with different proteins, such as plasminogen, S100A10, and HE4. It may be the complex interaction between these agents and Annexin A2 that plays a role in their malignant potential. Activation by phosphorylation appears to play a role in carcinogenesis, and to some extent, Annexin A2 appears to be regulated by Annexin A5. This accentuates the need to investigate the expression patterns of different Annexins within different forms of cancer. The ongoing regulation and collaboration of Annexin A2 could be the basis for its malignant potential and could be the focus for further investigation into targeting treatment for Annexins and plasminogen, S100, and HE4 proteins [61].

Thus, the in vitro effects of açaí seed oil in decreasing Annexin expression in colorectal adenocarcinoma cells and the result of molecular docking suggest a promising effect of this natural product in modulating the regulation of Annexin and, therefore decreasing its effects proliferative at the cellular level.

## 5. Conclusions

Considering the results obtained so far, açaí seed oil is a promising phytochemical product for the development of drug-active compounds with an anti-tumor effect. The use of the seed, a product that is usually discarded, will also contribute to sustainable development and stimulate the green economy of traditional communities, especially in the north and northeast regions. In vivo studies need to be performed in order to confirm these effects in colorectal cancer animal models.

## Figures and Tables

**Figure 1 metabolites-13-00789-f001:**
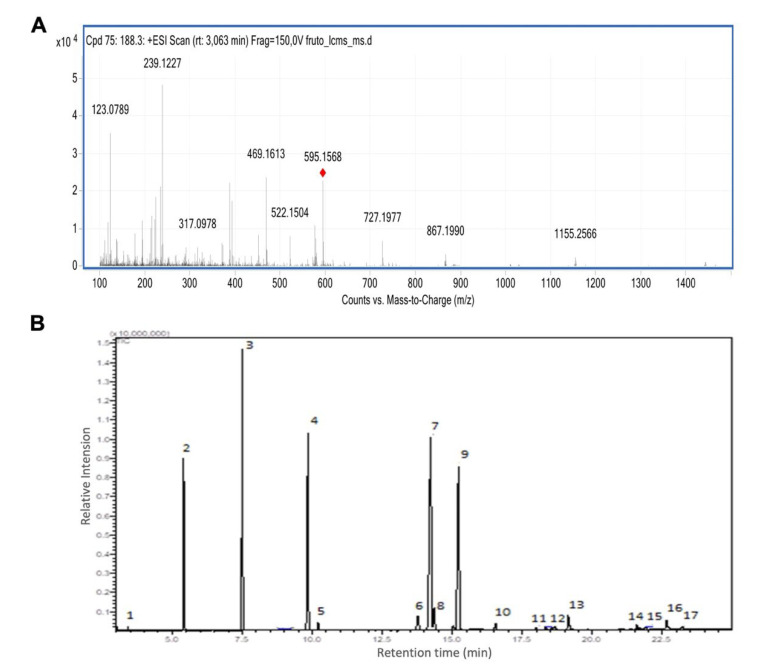
Açaí seed extract and oil characterization. (**A**) Mass spectrum of [M+H]^+^ of kaempferol-3-O-rutinoside. (**B**) Chromatogram of esters from fatty acids in seed oil from *Euterpe oleracea* Mart.

**Figure 2 metabolites-13-00789-f002:**
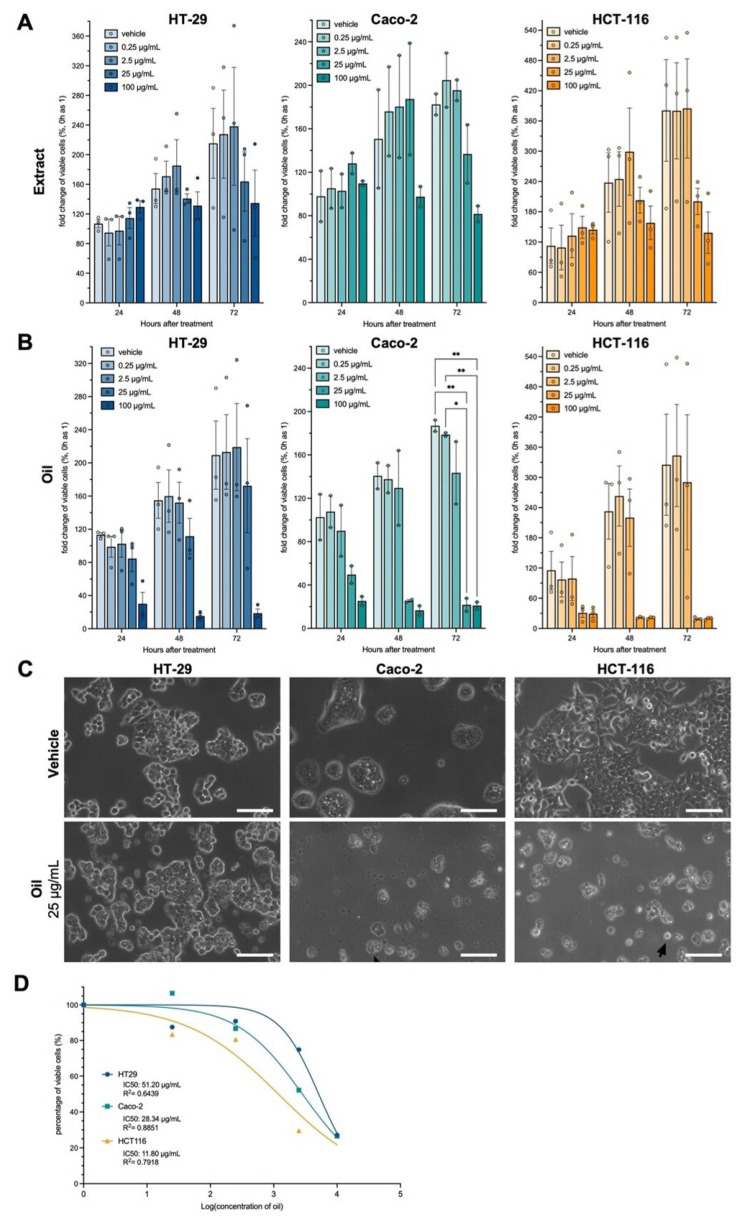
Açaí seed oil is cytotoxic in a cell line-specific manner. (**A**) Viability measurement in an MTT assay of three cell lines treated with açaí seed extract. (**B**) Viability measurement in an MTT assay of three cell lines treated with açaí seed oil. (**C**) Phase contrast micrographs of colon adenocarcinoma cell lines treated for 48 h with 25 µg/mL of açaí seed oil. (**D**) IC50 determination for each cell line treated for 24 h with açaí seed oil. Two-way ANOVA followed by Tukey post-hoc. Sphericity was assessed and Greenhouse–Geiser correction used when necessary. * *p* < 0.05; ** *p* < 0.01. Scalebar = 20 µm.

**Figure 3 metabolites-13-00789-f003:**
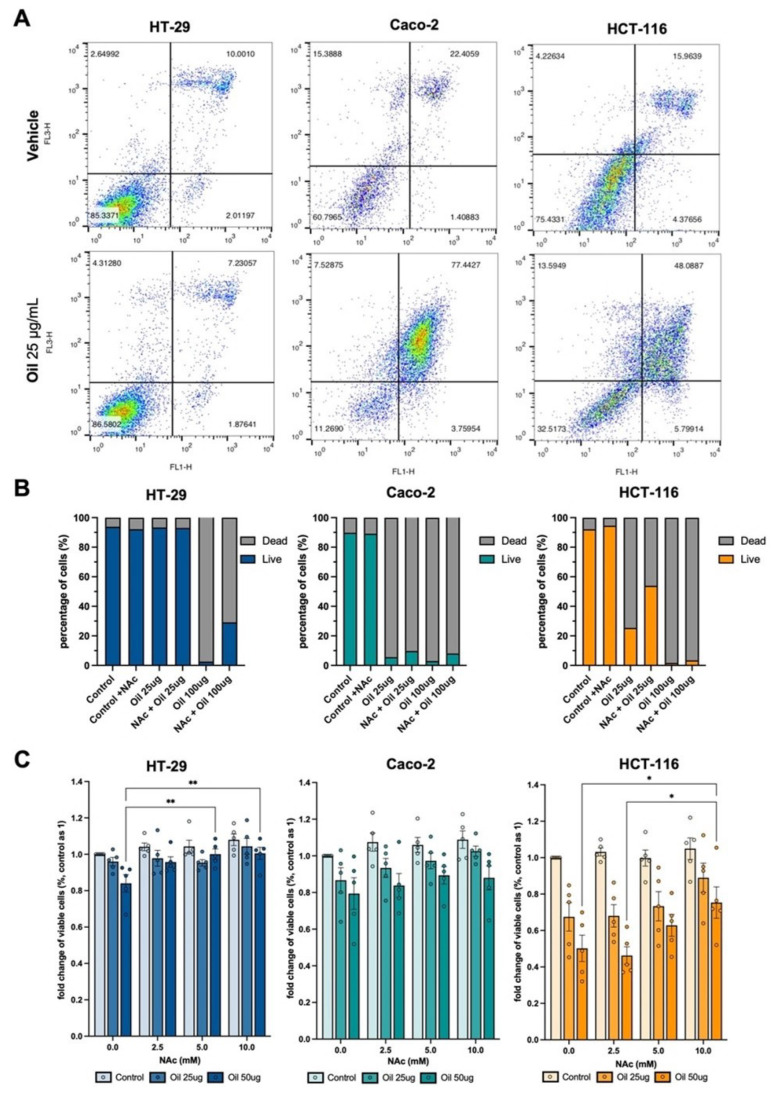
Açaí seed oil induces cell death, which can be partially reverted through the inhibition of reactive oxygen species. (**A**) Annexin V/PI assay read in the flow cytometer indicating increased apoptotic death in Caco-2 and HCT-116 cell lines 24 h after 25 µg/mL açaí seed oil treatment. (**B**) Trypan blue assessment of live/dead cells through manual counting of cells pre-treated with 5 mM NAc for 2 h and then exposed to 25 or 100 µg/mL açaí seed oil. (**C**) Viability measurement in an MTT assay of three cell lines treated with açaí seed oil and a gradient of NAc (0–2.5–5–10) to evaluate ROS impact over oil-induced death. Two-way ANOVA followed by Tukey post-hoc. Sphericity was assessed and Greenhouse–Geiser correction used when necessary. * *p* < 0.05; ** *p* < 0.01.

**Figure 4 metabolites-13-00789-f004:**
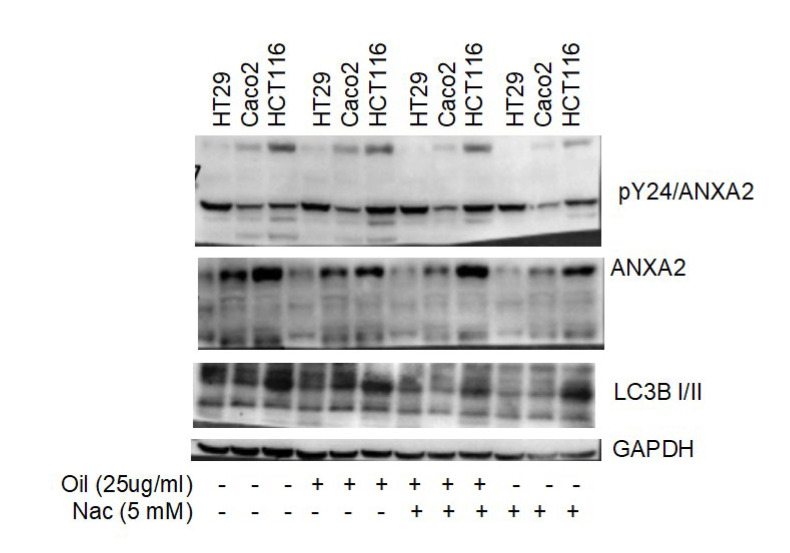
LC3B and ANXA2 protein levels are related to ROS-mediated apoptosis after açaí seed oil treatment. Analysis of total and Y24-phosphorylated ANXA2 and LC3B protein through western blotting. Cells were pre-treated with 5 mM of NAc and treated with 25 ug/mL (with their respective controls). GAPDH was used as housekeeping.

**Figure 5 metabolites-13-00789-f005:**
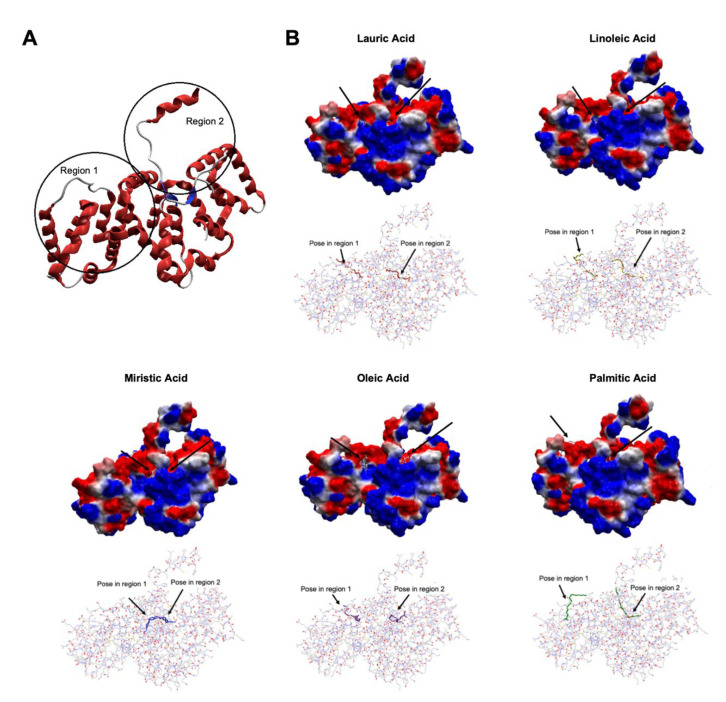
ANXA2 molecular docking analysis with fatty acids. (**A**) Preferential regions of ANXA2 for the binding of the fatty acids: palmitic, myristic, lauric, oleic, and linoleic according to the blind docking. (**B**) Poses selected by docking for the MD simulations. ANXA2 is represented in both electrostatic surface (right) and wire (left) representations. The arrows point to the positions of the poses in Regions 1 and 2.

**Figure 6 metabolites-13-00789-f006:**
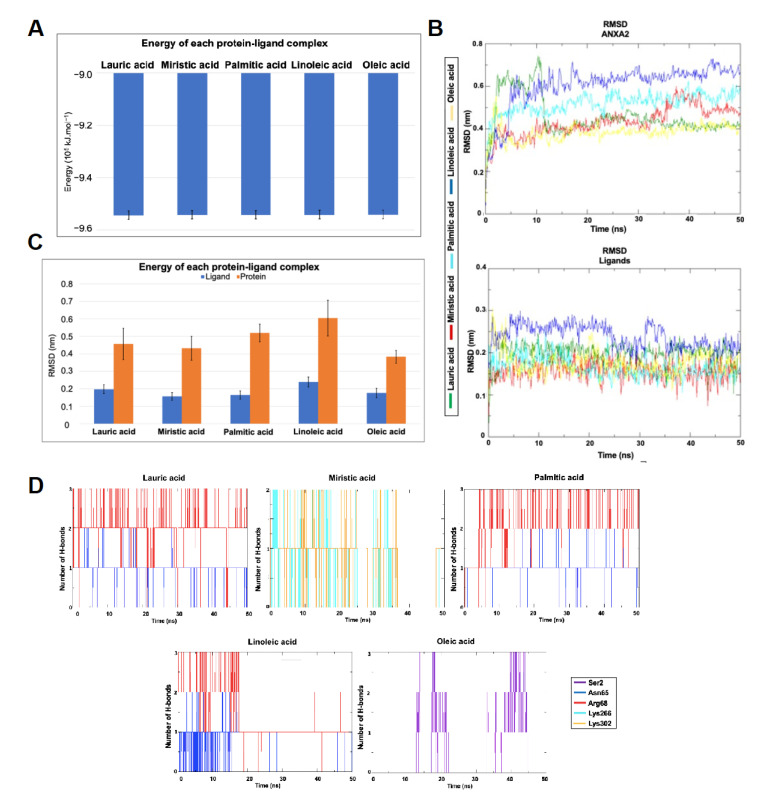
Molecular dynamic simulations of ANXA2 docking. (**A**) Average and standard deviation of the total energy of the ANXA2–ligand complexes during the MD simulations. (**B**) RMSD plot for the complexes ANXA2–ligand during the 50 ns of MD simulation. (**C**) Average and standard deviation of RMSD for the complexes ANXA2–ligand during the MD simulations. (**D**) H-bonds observed between ANXA2 and the ligands during the MD simulation.

**Table 1 metabolites-13-00789-t001:** Esters from fatty acids founded in *Euterpe oleracea* Mart. seed oil.

Fatty Acids	Peaks	Carbon Chain	% Area— Retention Time (min)	Retention Time (min)
Capric acid	1	C 10:0	0.13	3.429
Lauric acid	2	C 12:0	8.9	5.407
Myristic acid	3	C 14:0	18.03	7.508
Palmitic acid	4	C 16:0	16.61	9.845
Palmitoleic acid	5	C 16:1	0.58	10.206
Stearic acid	6	C 18:0	1.81	13.765
Oleic acid	7	C 18:1	26.94	14.228
Acid (isomer)	8	C 18:1	2.21	14.438
Linoleic acid	9	C 18:2	19.92	15.223
Linlenic acid	10	C 18:3	0.62	16.538
Nanodecenoic acid	11	C 19:0	0.21	17.985
Eicosenoic acid	12	C 20:0	0.35	18.326
Non-esterified myristic acid	13	C 14:0	1.75	19.134
Behenic acid	14	C 22:0	0.43	21.586
Eicosatrienoic acid	15	C 23:0	0.22	21,904
Non-esterified palmitic acid	16	C 16:0	0.88	22.649
Tricosanotrienoic acid	17	C 23:0	0.22	23.212

**Table 2 metabolites-13-00789-t002:** Energy values and interaction residues of the fatty acids in the regions 1 and 2 of ANXA2.

Ligand	Region	MolDock Score (kcal/mol)	Interacting Residues
Lauric acid	1	−99.42	Arg179
	2	−84.21	Asn65, Arg68, Ser296
Myristic acid	1	−94.63	Asn65, Arg68
	2	−88.14	Ser22, Lys302
Palmitic acid	1	−93.35	Asp187, His224, Lys227
	2	−99.54	Asn65
Oleic acid	1	−111.42	Arg178
	2	−97.39	Ala29
Linoleic acid	1	−105.94	Arg179
	2	−112.26	Asn65, Arg65, Ser296

## Data Availability

All data are included in the manuscript.

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
