# Peer review of "Açaí (Euterpe oleracea Mart.) Seed Oil Exerts a Cytotoxic Role over Colorectal Cancer Cells: Insights of Annexin A2 Regulation and Molecular Modeling"

_metabolites, 2023, doi:10.3390/metabo13070789_

Round 1
Reviewer 1 Report
This study aims to investigate the in vitro effects of açaí seed oil on three different colorectal adenocarcinoma cells (CACO-2, HCT-116, and 30 HT-29 cell lines). The authors treated the cell lines with açaí seed oil for 24, 48, and 72 h and analysed cell viability, death, and morphology. They also performed molecular docking simulations to evaluate the interaction between the major compounds in açaí seed oil and Annexin A2. It is an article with several methods. It is well written and concluded. Therefore this study can be published after following corrections:
1. The article contains some typing mistakes. The authors must carefully read the article and make necessary improvements (one example: line 86 Despite must be written as despite.
2. Line 98 should contain the names of the cell lines
3. Line 194 what is the parenthesis stand for in that line “()”?
4. The sentence in line 307-308 is unclear. Please fix it.
5. “Antioxidants, both natural and synthetic, are increasingly used as functional food additives and nutritional supplements. Oxidative reactions can damage proteins, lipids, and DNA due to an uneven ratio of antioxidants to free radicals. The capacity of antioxidants to stop oxidative deterioration in food and pharmaceutical items, as well as in the body and against disease processes brought on by oxidative stress, has heightened interest in them. Numerous examinations exploring the connection between food and human health have been carried out using diverse study focus samples. [1-2]”.
Zengin, G., Mahomoodally, M. F., Aktumsek, A., JekÅ‘, J., Cziáky, Z., Rodrigues, M. J., ... & Picot-Allain, C. (2021). Chemical profiling and biological evaluation of Nepeta baytopii extracts and essential oil: An endemic plant from Turkey. Plants, 10(6), 1176.
Gulcin, İ. (2020). Antioxidants and antioxidant methods: An updated overview. Archives of toxicology, 94(3), 651-715.
Author Response
This study aims to investigate the in vitro effects of açaí seed oil on three different colorectal adenocarcinoma cells (CACO-2, HCT-116, and 30 HT-29 cell lines). The authors treated the cell lines with açaí seed oil for 24, 48, and 72 h and analysed cell viability, death, and morphology. They also performed molecular docking simulations to evaluate the interaction between the major compounds in açaí seed oil and Annexin A2. It is an article with several methods. It is well written and concluded. Therefore this study can be published after following corrections:
Initiatilly, we would like to
- The article contains some typing mistakes. The authors must carefully read the article and make necessary improvements (one example: line 86 Despite must be written as despite.
Answer: We fix it.
2. Line 98 should contain the names of the cell lines
Answer: We included cell lines names.
3. Line 194 what is the parenthesis stand for in that line “()”?
Answer: We included this information (DMSO) in parenthesis.
4. The sentence in line 307-308 is unclear. Please fix it.
Answer: We fix it
5. “Antioxidants, both natural and synthetic, are increasingly used as functional food additives and nutritional supplements. Oxidative reactions can damage proteins, lipids, and DNA due to an uneven ratio of antioxidants to free radicals. The capacity of antioxidants to stop oxidative deterioration in food and pharmaceutical items, as well as in the body and against disease processes brought on by oxidative stress, has heightened interest in them. Numerous examinations exploring the connection between food and human health have been carried out using diverse study focus samples. [1-2]”.
Zengin, G., Mahomoodally, M. F., Aktumsek, A., JekÅ‘, J., Cziáky, Z., Rodrigues, M. J., ... & Picot-Allain, C. (2021). Chemical profiling and biological evaluation of Nepeta baytopii extracts and essential oil: An endemic plant from Turkey. Plants, 10(6), 1176.
Gulcin, İ. (2020). Antioxidants and antioxidant methods: An updated overview. Archives of toxicology, 94(3), 651-715.
Answer: We included this statement in the discussion.
Reviewer 2 Report
In this work, authors demonstrate (combining in vitro and in silico assay) that açaí seed oil has antitumor potential in colorectal adenocarcinoma cells. Since this paper could be of interest to the scientific community, I'd recommend publishing it in "Metabolites" after some minor revisions of the manuscript:
1. on pg.13, line 408, and pg.14, lines 413 and 414: “e” should probably be replaced with “and”. I suggest the authors check the entire manuscript to ensure this is corrected.
2. Check the manuscript for missing spaces, symbols, and super/subscripts. For example pg. 3, lines 119, 125, 138, 143-146, pg 9, lines 333-335.
3. I recommend editing the English language.
4. What does the „Peaks“ column in Table 1 represent?
5. Check the title of graph C in Figure 4! On the y-axis is rmsd (nm).
Author Response
In this work, authors demonstrate (combining in vitro and in silico assay) that açaí seed oil has antitumor potential in colorectal adenocarcinoma cells. Since this paper could be of interest to the scientific community, I'd recommend publishing it in "Metabolites" after some minor revisions of the manuscript:
- on pg.13, line 408, and pg.14, lines 413 and 414: “e” should probably be replaced with “and”. I suggest the authors check the entire manuscript to ensure this is corrected.
Answer: We fix it
2. Check the manuscript for missing spaces, symbols, and super/subscripts. For example pg. 3, lines 119, 125, 138, 143-146, pg 9, lines 333-335.
Answer: We fix it.
3. I recommend editing the English language.
Answer: We edited English language
4. What does the „Peaks“ column in Table 1 represent?
Answer: Peaks represent the açaí seed oil chemical compounds that are described in Figure 1B
5. Check the title of graph C in Figure 4! On the y-axis is rmsd (nm).
Answer: RMSD is root mean square deviation. This information is on page 6 line 296.
Reviewer 3 Report
1. What is the main question addressed by the research? The research was to analyze the effects of açaí seed oil on 29 colorectal adenocarcinoma (ADC) cells.
2. Do you consider the topic original or relevant in the field? Does it address a specific gap in the field? The presentation in this article expands on the possibilities of application and use of acai oil seeds.
3. What does it add to the subject area compared with other published material? There are no comments on this item
4. What specific improvements should the authors consider regarding the methodology? What further controls should be considered? There are no comments on this item.
5. Are the conclusions consistent with the evidence and arguments presented and do they address the main question posed? The conclusions could be strengthened by the resultant part on the basis of the studies obtained.
6. Are the references appropriate? Yes, links are appropriate. Literature corresponds to the stated topic of the article.
7. Please include any additional comments on the tables and figures. – Explanations of figures should be on one page. More clear.
Author Response
What is the main question addressed by the research? The research was to analyze the effects of açaí seed oil on 29 colorectal adenocarcinoma (ADC) cells
Answer: In this research we evaluated for the first time the effects of açaí seed oil on colorectal adenocarcinoma cell lines: CACO-2, HT-29 and HCT-116. We suggest that the cytotoxic effect of açaí seed oil is mediated by reactive oxygen species and annexin A2 is probably involved in this mechanism.
2. Do you consider the topic original or relevant in the field? Does it address a specific gap in the field? The presentation in this article expands on the possibilities of application and use of acai oil seeds.
Answer: The topic is original considering that is the first article to describe the effects of açaí seed oil on colorectal cancer cell lines, addressing the mechanism of action.
3. What does it add to the subject area compared with other published material? There are no comments on this item
Answer: There are no manuscripts addressing the effects of açaí seed oil on cancer. This is the first article using this natural product. Natural products are good sources of anticancer drugs.
4. What specific improvements should the authors consider regarding the methodology? What further controls should be considered? There are no comments on this item.
Answer: This study addresses the açaí seed oil effects on colorectal cancer. The study suggests that açaí seed oil interacts with ANNEXIN A2. We will perform tests with animal in order to confirm these effects.
5. Are the conclusions consistent with the evidence and arguments presented and do they address the main question posed? The conclusions could be strengthened by the resultant part on the basis of the studies obtained.
Answer: Our results are extremely consistent with the used methodology and the objectives of the study. The conclusions are clearly and do not extrapolate the results presented.
6. Are the references appropriate? Yes, links are appropriate. Literature corresponds to the stated topic of the article.
Answer: The references are appropriate for the manuscript.
7. Please include any additional comments on the tables and figures. – Explanations of figures should be on one page. More clear.
Answer: We fix it.